# Position: LLMs can't jump

**Tom Zahavy** [1]

## Abstract

How do we fundamentally discover new things? In a letter to Maurice Solovine, Albert Einstein conceptualized discovery as a cyclical process involving an intuitive 'jump' from sensory experience to axioms, followed by logical deduction. While Generative AI has mastered Induction (statistical pattern matching) and is rapidly conquering Deduction (formal proof), we argue it lacks the mechanism for Abduction—the generation of novel explanatory hypotheses. Using Einstein's formulation of General Relativity as a computational case study, we demonstrate that the prevailing theory of "creativity as data compression" (induction) fails to account for discoveries where observational data is scarce. This position paper argues that while a modern Large Language Model could plausibly execute the deductive phase of proving theorems from established premises, it is structurally incapable of the abductive 'Jump' required to formulate those premises. We identify the translation of simulation into formal axioms as the critical bottleneck in artificial scientific invention, and propose that physically consistent, multimodal world models offer the necessary sensory grounding to bridge this divide.

## 1. Introduction

What characterizes the cognitive leap required for scientific invention? A prevailing view in the AI community, notably championed by Schmidhuber (2008), suggests that scientific discovery is fundamentally a problem of compression—the search for a simple program that concisely explains observations. This view implicitly frames discovery as *Induction*: inferring general rules from observations based on statistical frequency. Concurrently, the success of systems like AlphaProof (Hubert et al., 2025) in Olympiad-level mathe-

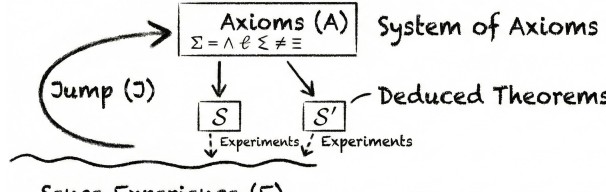

*Figure 1.* A generative AI reconstruction of Einstein's E-J-A diagram. Einstein drew this diagram in a letter to Maurice Solovine, showing a cyclical line jumping from Sense Experience (E) to Axioms (A) via a Jump (J), and then deducing logical consequences. Ironically, the hallucination of the axiomatic symbols highlights the very difficulty of automating the jump.

matics suggests that AI is mastering *Deduction*: the formal derivation of theorems from established premises.

If scientific discovery were merely the sum of these two parts, modern Large Language Models (LLMs) should theoretically be capable of inventing theories like General Relativity given sufficient compute.

In this paper, we challenge this reductionist premise by treating Albert Einstein's formulation of General Relativity as a computational case study. Adopting Einstein's cyclical model of invention—illustrated in Figure 1—we map the process from Sense Experience ($E$) to a System of Axioms ($A$) via a conceptual Jump ($J$), followed by the deduction and verification of theorems. While we concede that a modern LLM could plausibly perform the deductive work if initialized with Einstein's assumptions, the formulation of the axioms remains the bottleneck.

By reconstructing the historical context in Section 2, we show that the scarcity of experimental data precludes induction as an explanation. Since axioms also cannot be deduced (being premises), we propose that scientific discovery requires a cognitive mechanism beyond induction and deduction: Abduction.

To formalize this, we adopt the framework of Peirce (1934), which categorizes inference based on the structural permutation of a *Rule* (function definition), a *Case* (input), and a *Result* (return value):

[1]Google DeepMind, London, United Kingdom. Correspondence to: Tom Zahavy <tomzahavy@gmail.com>.

*Proceedings of the 43$^{rd}$ International Conference on Machine Learning*, Seoul, South Korea. PMLR 306, 2026. Copyright 2026 by the author(s).

- **Deduction** (Rule + Case → Result) is the analytic application of a Rule to a Case to predict a Result. It is the only mode that guarantees truth (e.g., executing code to verify output).

- **Induction** (Case + Result → Rule) is the synthetic derivation of a Rule from the accumulation of Cases and Results. It validates hypotheses through statistical frequency (e.g., generating a function to satisfy unit tests).

- **Abduction** (Rule + Result → Case) is the inference of a Case (or a new Rule) to explain a surprising Result.

Unlike deduction, which guarantees truth, or induction, which finds pattern that generalize in data, abduction is a creative leap that invents a cause for a singular phenomenon. Crucially, Einstein achieved this via embodied simulation—using thought experiments to ground abstract symbols in physical sensation—enabling him to formulate axioms where no symbolic data previously existed.

**We argue that while Large Language Models have mastered the inductive compression of data and the deductive verification of theorems, they are structurally incapable of the abductive 'jump' required for scientific invention. We posit that this creative leap demands not just better language processing, but the integration of physically consistent World Models that ground abstract symbols in sensory simulation.**

## 2. Background

### 2.1. Mechanics

In the 19th century, mechanics was regarded as the foundation of all physics. Through the lens of partial differential equations, scientists could explain a vast array of phenomena: the propagation of sound, hydrodynamics, the motion of discrete masses, and even the kinetic theory of gases (linking viscosity, heat conduction, and diffusion). At the time, even light was understood through this mechanical framework, described as a wave moving through the ether.

Yet, the mechanical worldview began to fracture. Through the contributions of Maxwell, Faraday, Hertz, and Mach, the laws of electromagnetism were unified into Maxwell's equations. Newtonian mechanics struggled to explain these electromagnetic fields, signaling the end of mechanics as the sole governing paradigm of physics. Physics found itself divided into two conceptual elements: material points with forces at a distance between them and continuous fields. Einstein found this division unacceptable and was driven to create a field theory for gravity that would replace the old idea of action at a distance.

Meanwhile, a crisis was brewing regarding the nature of

light. Because light behaves as a wave, scientists assumed it traveled through a medium they called the ether. However, the famous Michelson-Morley experiment in the late 19th century shattered this assumption. They attempted to measure Earth's velocity relative to the ether but failed to do so. Even more shocking was the observation that the speed of light did not vary with the Earth's movement around the Sun. Attempts to salvage the ether theory resulted in increasingly complex and artificial explanations, such as ether wind, all of which ultimately proved futile.

In addition, Newton's theory of gravitation was incredibly robust, accurate to an astonishingly small margin of error. Newton confirmed Galileo's discovery that all bodies fall at the same speed regardless of mass by performing pendulum experiments. In particular we have, $F_{grav} = m_i \frac{d^2x}{dt^2} = m_g g$, so if $m_i = m_g$ we have that the acceleration is constant $\frac{d^2x}{dt^2} = g$ and independent of mass. Newton's experiments validated that $\frac{m_i}{m_g} = 1$ with an accuracy of $10^{-3}$. Over the centuries, this precision was refined even further—Laplace achieved $10^{-7}$ and Eötvös reached $10^{-9}$.

In fact, there was only one known anomaly: a tiny shift in Mercury's orbit known as the advance of perihelion (Leverrier 1845). Scientists were so confident in Newton's laws that they didn't question the theory; instead, they hypothesized that an undiscovered planet, dubbed 'Vulcan,' was hiding near the Sun and causing the disturbance.

### 2.2. Special relativity

In 1905, Einstein resolved the contradictions of the Michelson-Morley experiment in a way that fully aligned with Maxwell's equations. He founded his new theory on two key postulates. *Principle of relativity:* The laws of physics are identical in all inertial frames of reference. *Invariance of the speed of light:* The speed of light in a vacuum, $c$, is constant in all inertial frames of reference.

The Michelson-Morely experiment was designed to detect Earth's movement through a hypothetical ether, and found that there is no change in light speed; Light always travels at $c$ so its speed doesn't change relative to a moving Earth, exactly the second postulate.

From the two postulates, Einstein derived the Lorentz transformation, which relates the coordinates of a rest frame to one moving at a constant relative velocity $v$. The resulting transformation for time is:

$$t' = \frac{t - \frac{v}{c^2}x}{\sqrt{1 - \frac{v^2}{c^2}}}. \tag{1}$$

Historically, predecessors like Poincaré referred to the variable $t'$ as 'fictitious time'. However, Einstein's interpreta-

tion was radical. He argued that $t'$ was not a calculation artifact, but "time plain and simple". To demonstrate this, he introduced the concept of time dilation: if two originally synchronized clocks are separated and one undergoes motion at velocity $v$, they will no longer report the same time upon clearer reunification. With this insight, Einstein shattered the Newtonian paradigm of absolute, universal time, replacing it with a temporal reality that is local to every observer.

## 2.3. General relativity

Einstein's new theory was intrinsically limited to **inertial** frames—observers moving at constant velocities without acceleration. This specific constraint is the origin of the name 'special' relativity. Motivated by the earlier work of Ernst Mach, Einstein was convinced that inertial frames should hold no privileged status. Consequently, he sought a generalization of the theory applicable to any frame of reference, embarking on the quest for **General** relativity. This seven-year odyssey was characterized by profound physical hunches, vivid thought experiments, and rigorous mathematical formalization, interspersed with periods of exhaustion and error. In the following analysis, we adopt the framework of Norton (2020), examining Einstein's progress through three distinct phases.

**Ideation (1907-1912).** Einstein took his first concrete steps toward General Relativity in 1907, when Johannes Stark commissioned him to write a comprehensive review of relativity. The task initially seemed straightforward: Einstein needed to examine established branches of physics to ensure they fit within the new framework of space and time he had proposed in 1905.

The work progressed smoothly. Electrodynamics required no changes, as it was already compatible with the Lorentz transformation. Mechanics needed some adjustment, specifically regarding energy, momentum, and mass, which led Einstein to formalize the equivalence of mass and energy ($E = mc^2$). He even sketched out a relativistic treatment for thermodynamics.

However, as he finalized the review, Einstein felt a compelling need to go further. He wanted to generalize the principle of relativity to include not just constant motion, but accelerated motion. He was struck by a profound insight—later calling it his 'happiest thought'—that acceleration mimics gravity, suggesting that inertia itself is a gravitational effect. These ideas culminated in his 1912 theory of static gravitational fields, where he boldly proposed that gravity bends light, slows down clocks, and that the speed of light is not constant, but varies depending on the gravitational potential.

**Consolidation (1912-1913).** The pivotal transition toward General Relativity occurred between the summer of 1912 and early 1913. Struggling to translate his physical intuition into a rigorous theory, Einstein realized that the mathematics of curvature was the key. To master this complex field, he turned to his friend and mathematician, Marcel Grossmann, in Zurich. Their collaboration was documented in the famous "Zurich Notebook" and culminated in the 1913 paper known as the Entwurf ("Sketch").

Einstein consolidated a set of physical requirements and conceptual pillars that he intended the new theory to satisfy (see Section A for more details):

1. Generalized Relativity Principle: Extension of special relativity to accelerated frames

2. Equivalence Principle: Indistinguishability of gravity and acceleration

3. Geodesic Principle: The motion of free-falling bodies in spacetime

4. "Gravity Gravitates": Gravitational energy itself acts as a source

5. Stress-Energy Tensor: The source of the gravitational field

6. Generalized Poisson Equation: The field equation structure

7. Newtonian Limit: Recovery of classical gravity

**The Fatal Error.** In the mathematical section, Grossmann came agonizingly close to the final answer. He identified the Riemann curvature tensor as the correct measure for spacetime curvature. He even contracted this tensor to derive a quantity ($G_{ik}$) that is nearly identical to the modern Einstein tensor. From a modern perspective, the finish line was in sight. But despite all of this, they stopped short. The new equations had to pass a crucial test: they needed to reproduce Newton's simple law of gravity in weak, static fields (Principle 7). In a fatal error, Grossmann concluded that their candidate tensor did not reduce to the Newtonian expression. However, as they only figured out later, the error lay not in the geometry, but in the assumption about the static field itself.

Believing this path was blocked, they abandoned it. Einstein was forced to construct a set of field equations based purely on physical clues, such as conservation laws and his earlier work on static fields. The result was a mess: instead of one simple Newtonian equation, he produced ten complicated, non-linear equations with no clear geometric meaning—a detour that would delay the final theory for two more years.

**Mathematical validation (1913-1915).** The years 1913 to 1915 were defined by a grueling struggle to correct and perfect the 1913 draft. With the publication of the 'Entwurf' paper in mid-1913, Einstein initially believed the heavy lifting was done and only details remained. This feeling was short-lived. As months turned into years, he found himself working harder and harder to justify a theory that was misshapen.

By the summer of 1915, the evidence against his old theory was mounting. He knew it failed to explain the anomalous orbit of Mercury. He discovered it could not account for rotational motion. Finally, he realized that his sophisticated attempts to prove the theory's uniqueness in late 1914 were flawed. In a state of mounting desperation, Einstein abandoned the 'adapted' coordinate systems of the 'Entwurf' and returned to his earlier intuition from 1912: the theory needed to work in all coordinate systems.

What ensued was perhaps the most intense month of Einstein's career. Spurred by the knowledge that the renowned mathematician David Hilbert was racing to solve the same problem, Einstein entered a frenzy of productivity, submitting a new paper to the Prussian Academy every week for four consecutive weeks. His first communication on November 4 proposed a solution, yet errors persisted. By November 11, he had refined the theory but difficulties remained; however, on November 18, he announced the thrilling result that his evolving equations correctly predicted the anomalous orbit of Mercury. Finally, on November 25, the fourth communication unveiled the equations of General Relativity:

$$R_{ik} - \frac{1}{2}g_{ik}R = -\kappa T_{ik}. \tag{2}$$

Here, $R_{ik}$ is the Ricci curvature tensor, $R$ is the Ricci scalar, $T_{ik}$ is the Stress-Energy tensor, and $g_{ik}$ is the metric tensor. The expression on the left represents the geometry of spacetime (curvature) as determined by the metric, while the expression on the right represents the matter and energy.

## 3. Alternative Views

### 3.1. The Limits of Inductive Inference

"One not infrequently hears the viewpoint expressed that physicists are merely noticing patterns... It seems to me, however, that such a viewpoint is extraordinarily wide of its mark... When Einstein's theory was first put forward, there was really no need for it on observational grounds. ...Einstein was not just 'noticing patterns' in the behavior of physical objects. He was uncovering profound mathematical structure that was already hidden in the very working of the world."

*– Roger Penrose*

This distinction between noticing patterns and uncovering structure highlights the boundary between AI as it exists today and the AI required for scientific invention. The prevailing view in machine learning aligns with the "Theory of Compression Progress," (Schmidhuber, 2008) which posits that scientific discovery is driven by the inductive desire to compress data. In this framework, the "joy" of discovery is the rate at which complex observations become subjectively simpler through better prediction. This inductive approach has yielded impressive results in data rich environments: sparse optimization has successfully extracted partial differential equations from data (Schaeffer, 2017), and the "AI Physicist" (Wu & Tegmark, 2019) successfully rediscovered conservation laws from simulated trajectories.

However, we argue that this inductive framework is insufficient to explain the invention of General Relativity. While Einstein sought logical simplicity, his process was not driven by data compression—primarily because there was no statistically significant supervised training set to compress.

At the time of invention, Newtonian gravity faced no empirical crisis. The equivalence of inertial and gravitational mass had been verified to a precision of $10^{-9}$, and Newton's laws were accurate to an astonishingly small margin of error. The only known anomaly—the advance of Mercury's perihelion—was viewed not as a failure, but as evidence of a hidden variable: the undiscovered planet "Vulcan".

This highlights the fundamental limitation of "creativity as compression": scientific invention often occurs in the absence of a supervised error signal. An AI operating as an inductive optimization engine would have found the Newtonian loss function to be near-zero. Without a significant discrepancy between prediction and observation, there is no gradient to drive the system toward a foundational restructuring of spacetime.

If modern Transformer models struggle to reverse-engineer basic arithmetic rules (Gambardella et al., 2024; Yang et al., 2024), it is difficult to see how it could invent a new physics in the absence of massive datasets. Furthermore, even when data is available, inductive systems risk converging on heuristic shortcuts rather than causal laws. Vafa et al. (2025) demonstrate that without the correct inductive bias, foundation models often discover flawed world models that satisfy the data but fail to capture the underlying structure. The empirical evidence required to validate General Relativity—from the Eddington experiment to relativistic GPS corrections—arrived after the theory was formulated. Einstein was not compressing a noisy dataset to fit a regression curve; he was constructing a logical framework to uncover a physical structure that the data had not yet revealed.

Lastly, it could be argued that while Einstein was not compressing data, he was compressing the hypothesis space by

seeking to unify the laws of inertia and gravity into a single framework (Minimum Description Length). However, logical simplicity is often a retrospective property. While the final theory of General Relativity is elegant, the search path to get there was paved with complexity, abandoned tensors, and incorrect equations. A compression-driven AI might prefer to patch Newtonian gravity with a parameter like the 'Vulcan' planet hypothesis rather than expanding the hypothesis space to include non-Euclidean geometry, which increases complexity before it simplifies it.

## 3.2. The Limits of Deduction

"I see on one side the totality of sense experiences, and on the other, the totality of the concepts and propositions that are laid down in books. The relations between concepts and propositions among themselves are of a logical nature... The concepts and propositions get 'meaning', or 'content', only through their connection with sense experiences."

*– Albert Einstein*

Einstein explicitly distinguished between the domain of sensory experience and the domain of logical processing. In our framework, this latter domain corresponds to Deduction $(A \rightarrow S)$: the rigorous derivation of theorems from a fixed set of axioms.

Even the motivation to begin the search for General Relativity contained a strong deductive component. Einstein's drive was not sparked by data anomalies. There was no "error signal" in the Newtonian observation history, but by a conceptual inconsistency: the clash between mechanical action at a distance and the emerging field theories of electromagnetism. While modern LLMs will struggle to find such an idea due to the "weak signal" (there was no requirement to replace Newton's gravity), the structural task of proposing a field theory for gravity by mimicking Maxwell's equations is fundamentally a deduction operation.

It is plausible that a modern AI, optimized to search for inconsistencies in scientific literature, could identify this contradiction. Much like a system identifying "buggy code," an AI could flag that the constant speed of light in Maxwell's equations is incompatible with Newtonian absolute time. However, identifying the error is distinct from generating the fix. While the structural task of proposing a field theory for gravity is a deductive operation, selecting the correct axioms to resolve the conflict requires more than logical consistency.

The period between 1913 and 1915 illustrates this deductive struggle. It was defined not by flashes of insight, but by a grueling, mechanical search to identify the correct mathematical framework to satisfy Einstein's postulates.

This phase closely mirrors the capabilities of modern neuro-symbolic AI. Einstein's collaboration with Marcel Grossmann was essentially a "search" process over geometric constraints. Notably, they identified the Riemann curvature tensor as the correct object but discarded it due to a "fatal error"—the mistaken belief that it did not reduce to Newtonian gravity in static fields. It took two years of exhaustion to debug this assumption and produce the final field equations.

The landscape of mathematical discovery has been recently transformed by automation. Proof assistants based on dependent type theory, such as Lean, have matured into robust platforms supported by extensive libraries (mathlib Community, 2019). In 2024 LLMs have achieved remarkable fluency in proof generation: AlphaProof (Hubert et al., 2025) achieved silver-medal performance on IMO problems. Successors like Gemini, DeepSeekMathV2, and GPT-5 attained gold-level performance in 2025 and systems like Aristotle (Achim et al., 2025) produced verified solutions to open research questions.

"At the age of twelve I experienced a second wonder of totally different nature - in a little book dealing with Euclidean plan geometry, ..., were assertions, that could be proved with such certainty that any doubt appeared to be out of question. This lucidity and certainty made an indescribable impression on me."

*– Albert Einstein*

Given this trajectory, we posit that a modern LLM, initialized with the specific physical assumptions available to Einstein in 1915, could plausibly derive General Relativity. The derivation of the perihelion precession of Mercury, once the field equations are set, is a verifiable logical task $(A \rightarrow S)$. Furthermore, current systems are theoretically capable of identifying and eliminating erroneous constraints—such as Einstein's error regarding static fields—by systematically optimizing over subsets of axioms.

However, this capability comes with a critical caveat. An AI can deduce the consequences of "The Equivalence Principle" only if those concepts are provided as inputs. As Einstein noted, logical thinking is limited to connections between concepts; it cannot generate the concepts themselves from raw sensory data. The 1913 derivation failed not because the logic was flawed, but because the axioms were incorrect. This leads us to the fundamental bottleneck: what cognitive process allowed Einstein to generate the "Equivalence Principle" in the first place? To understand this, we must look beyond logic to the mechanism of the "Jump" $(J)$.

Finally, even if an AI possesses the deductive capacity to derive Einstein's equations from his postulates, a fundamental problem of intent remains. Unlike formal theorem

proving, where the goal is a specific open conjecture, Einstein was not trying to prove a theorem but to construct a predictive model of reality. While the anomalous perihelion of Mercury offered a verification target, it was not considered important enough. Crucially, the definitive validation—measuring the gravitational bending of starlight passing near the Sun by Eddington—arrived years after the theory was formulated. Deduction ($A \rightarrow S$) is strictly a downstream process: it unfolds the logical consequences of a theory, but it lacks both the upstream capacity to generate axioms and the external grounding to validate them.

## 4. Abduction: The missing Jump

"Then there occurred to me the happiest thought of my life... for an observer falling freely from the roof of a house there exists—at least in his immediate surroundings—no gravitational field... The observer therefore has the right to interpret his state as 'at rest.' Because of this idea, the uncommonly peculiar experimental law that in the gravitational field all bodies fall with the same acceleration attained at once a deep physical meaning."

*– Albert Einstein*

How does the mind formulate new axioms in the absence of sufficient data? Einstein's 'happiest thought' provides the answer: Manipulative Abduction (Magnani et al., 2009). This process relies on embodied simulation—an active interaction with mental models to generate hypotheses through thinking by doing, thereby accessing knowledge beyond the reach of pure deduction. Einstein did not bridge Special Relativity and gravitation by gathering observations, but by simulating the physical feelings of an observer inside a sealed environment.

To make this concept of manipulative abduction more accessible, we can look to a simpler, classical example: Archimedes' discovery of his buoyancy principle. Tasked

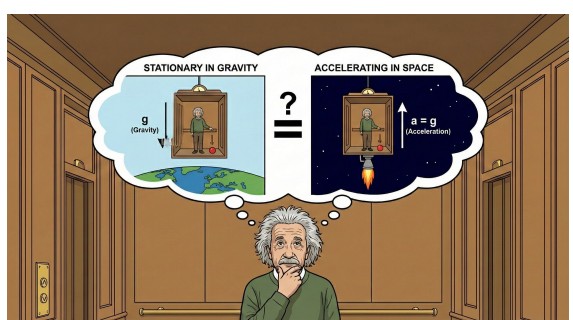

*Figure 2.* Einstein's thought experiment of the equivalence principle (AI generated).

with determining if a crown was made of pure gold without damaging it, Archimedes made his abductive leap not through mathematical derivation, but while stepping into a bath. By experiencing the bodily sensation of water rising and being displaced by his own volume, he performed an embodied sensory simulation. This raw physical feedback allowed him to bridge a sensory experience directly to a new axiomatic concept—volume displacement—which could not be deduced from the existing mathematical or linguistic theories of his era. Just as Archimedes used the physical sensation of displacement to formulate a new law, Einstein used the simulated sensation of gravity.

We conceptualize this thought experiment as a two-stage process. First, an observation is imagined via *simulation*. Second, an explanation is derived for that observation via *abductive* reasoning. Modern benchmarks like ARC-AGI (Chollet et al., 2025) already test the latter. In ARC, models must infer hidden rules from sparse examples (2–5 grid pairs). Since the data is too sparse for statistical induction and lacks the explicit instructions required for deduction, the solver must make an abductive leap to the most plausible explanation. However, as we argue next, while ARC captures the *logical* leap, it misses the *manipulative* component—the physical sensation and embodied simulation that drove Einstein's insight.

**Simulation as Physical Variation.** The first process—inventing a question to force progress—can be viewed through the lens of modern AI as Test Time Reinforcement Learning. This paradigm involves inventing new variations of a problem and learning to solve them, a strategy successfully applied to solve the Penrose position in chess (Zahavy et al., 2024) and to achieve silver medal standards in the IMO (Hubert et al., 2025).

However, a critical distinction remains. While symbolic variations in chess and mathematics are bounded by fixed rules (axioms), Einstein's variation required inventing new axioms based on a physical intuition that did not yet exist in the mathematics. He envisioned a physicist inside an elevator being uniformly accelerated through deep space (Figure 2). Inside this enclosure, the sensory experience reveals a specific pattern: when objects are released, the floor rushes up to meet them. To the physicist, the objects appear to fall with identical acceleration, regardless of composition. Thus, the simulation here was not a permutation of symbols, but a manipulation of perceptual experience.

**Abduction and the Physical Prior**. The second process is Abductive Reasoning: the inference to the best explanation. Unlike deduction, which guarantees truth from premises, abduction seeks the simplest, most likely cause for an observation.

In Einstein's scenario, the existing Newtonian framework

offered no satisfying explanation for his imagined observation. He faced a silence in the space of language—a lack of prior symbolic representation:

> "The words or the language, as they are written or spoken, do not seem to play any role in my mechanism of thought."

> – *Albert Einstein*

To fill this void, he relied on a physical prior. Because the simulated sensory experience of acceleration was indistinguishable from the remembered sensory experience of gravity, Einstein abducted that they must be the same phenomenon. The field inside the box was not a fake inertial effect; it was, by definition, a genuine gravitational field.

**From Chinese Rooms to World Models**. This cognitive process—anchoring abstract symbols in tangible physical simulations—is known as manipulative abduction (Magnani et al., 2009). This stands in sharp contrast to the operational mechanics of LLMs.

While LLMs excel at Induction (finding patterns in data), they lack the sensory agency required to ground these symbols in physical reality. They operate as high-dimensional "Chinese Rooms" (Harnad, 1990), manipulating the language of physics without access to the physical referents that give that language meaning. This limitation prevents the AI from making the Abductive Jump ($E \rightarrow A$). While Einstein could ground his axioms in the physical experience of a falling body, an LLM is confined to the logical deduction of existing texts.

It is worth addressing whether emerging techniques such as prompt engineering, in-context learning, or iterative context refinement might unlock abductive capabilities in LLMs. While these techniques significantly improve reasoning performance within an established symbolic space—enabling the model to navigate and recombine existing concepts more efficiently—they are fundamentally constrained by the boundaries of that symbolic space. Prompting cannot provide the sensory grounding required for manipulative abduction, which entails generating axioms without symbolic precedent. It cannot bridge the gap between abstract tokens and raw, un-symbolized physical experience.

This deficit in physical grounding is central to recent critiques of AI. Experts contend that despite linguistic mastery, current systems lack the spatial intelligence(Li, 2025) and internal world models(LeCun, 2022) required to reason about physical reality. Without the ability to perceive or interact with the world, LLMs struggle with spatial reasoning tasks that are trivial for toddlers .

The emergence of World Models offers a pathway to bridge this divide, but a critical distinction must be drawn between visual prediction and interactive simulation. Current video generation models like Veo exhibit intuitive physics(Hassabis, 2025) primarily as a byproduct of statistical correlation; they correctly generate a falling apple not because they model gravity, but because falling is the dominant continuation of unsupported object in their training distribution.

However, recent architectures like Genie (Bruce et al., 2024) mark a fundamental shift by introducing action-controllability into generative world models. Unlike passive video generators, Genie learns an action space that allows for agentic intervention—a prerequisite for Manipulative Abduction (thinking by doing). To replicate Einstein's elevator thought experiment, an AI cannot merely watch a video of an elevator; it must possess the capacity for counterfactual intervention (Pearl & Mackenzie, 2018). It must be able to essentially take control of the simulation to conceptually cut the cable.

We argue that physically consistent, action-controllable world models are not just useful, but are theoretically sufficient to enable abductive leaps. An interactive world model functions as a "synthetic laboratory" where an agent can run counterfactual simulations and test reasoning that goes beyond standard linguistic training data. Rather than merely predicting the next token or frame based on statistical correlation, the world model allows the agent to manipulate the environment, observe the consequences of novel actions, and abduce underlying rules. The sensory and interactive feedback from this synthetic laboratory serves as the grounding mechanism necessary to propose new axioms where no symbolic precedents exist in the language data. We propose that future iterations of such interactive environments, operating on a consistent latent physics manifold rather than just pixels, will provide the substrate necessary to transform the Abductive Jump from a mystical insight into a reproducible algorithmic process.

Lastly, its important to note that Einstein relied on his Physical Prior, using the sensation of gravity to prune the search space of possible axioms. However, manipulative abduction extends beyond physics. Historical scientific revolutions are often driven by strong, pre-symbolic intuitions—whether Kepler's Neoplatonic belief in the centrality of the Sun or the 'objective anger' that drove Marx's modeling of capital. To automate invention, we may need systems that do not just simulate the world, but hold strong beliefs or priors about how that world should be structured, using simulation to test those specific intuitions.

## 5. Conclusion

In this paper, we posed a fundamental question: Could a modern Artificial Intelligence, given the knowledge avail-

able to Einstein, invent General Relativity? Our investigation suggests that for current Large Language Models, the answer is no. While the field has successfully mechanized Induction (via statistical compression) and Deduction (via formal verification), these mechanisms alone are insufficient to sustain the cycle of scientific invention.

The prevailing Creativity as Compression hypothesis fails to account for this discovery because it presumes the existence of a pervasive error signal. Yet, the Newtonian paradigm faced no such crisis, and the data required to validate General Relativity did not exist until years after its formulation. Furthermore, while the deduction paradigm offers a path to derive field equations once axioms are set, it is ultimately a downstream process—a verification step within the invention loop, rather than the mechanism of invention.

This limitation is visible even in the apex of today's automated discovery systems. Agents such as Sakana's AI Scientist (Lu et al., 2024) and Google DeepMind's AlphaEvolve (Novikov et al., 2025) demonstrate the immense power of mechanizing scientific loops and evolutionary optimization. However, they highlight the very abductive gap we identify. The AI Scientist recombines existing symbolic concepts to optimize metrics—a sophisticated "Chinese Room" operation that lacks the sensory grounding to invent axioms without symbolic precedent. Similarly, while AlphaEvolve excels at optimization within a fixed framework, it relies on a gradient; Einstein, by contrast, had no error signal from Newtonian mechanics to drive his discovery. These systems lack the embodied world model required to perform the counterfactual physical simulations that drive the abductive Jump to entirely new paradigms.

Our analysis suggests that the critical bottleneck is this intuitive Jump from sensory experience to formal axioms ($E \to A$). We frame this work not as a definitive proof, but as establishing a theoretical framework to guide future empirical testing of abductive capabilities in generative systems. Einstein did not discover General Relativity by searching over symbols; he discovered it by simulating the sensual experience of a falling observer. The formulation of the Equivalence Principle was a self-contained act of physical abduction, where the premises were established solely through internal simulation, independent of immediate external verification.

To build an AI capable of true invention, we must therefore move beyond systems that merely read scientific literature to systems that can perceive the physical world. The emergence of physically consistent World Models offers a pathway to a synthetic laboratory. By enabling agents to run counterfactual simulations—to experience the physical consequences of a thought experiment—we may finally mechanize the feedback loop between intuition and logic.

Finally, we emphasize that this proposal is specifically tailored to the physical sciences, where the object of study is external material reality. In abstract domains such as Mathematics or Computer Science, the Sense Experience ($E$) may be grounded in high-dimensional topology or have other goals such as generality or minimality. While the necessity of the Abductive Jump remains universal, the nature of the simulation must be adapted to the ontology of the discipline: for physics, the substrate is the world; for mathematics, it is the abstract landscape of formal systems.

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

# A. Einstein's postulates of general relativity

**(1) Generalized Relativity:** While Special Relativity was restricted to inertial frames (those moving at constant relative velocities), Einstein sought to extend the principle of relativity to all frames of reference, including those in non-uniform, accelerated motion.

**(2) The Equivalence Principle:** Termed by Einstein as "the happiest thought of my life," this principle asserts that the local effects of a homogeneous gravitational field are physically equivalent to those of uniform acceleration in gravitation-free space. This thought experiment yielded immediate testable predictions, specifically the gravitational redshift of clocks and the deflection of light rays by massive bodies.

**(3) The Geodesic Principle:** This principle posits that free-falling objects traverse 'timelike geodesics'—the straightest possible trajectories within curved spacetime—thereby redefining gravity as a geometric phenomenon rather than a force. This generalizes the non-relativistic notion of a geodesic (such as the shortest path across a two-dimensional surface) into the four-dimensional framework of relativity.

**(4) "Gravity Gravitates":** By synthesizing the mass-energy equivalence ($E = mc^2$) with the equivalence of inertial and gravitational mass ($m_i = m_g$), Einstein deduced that the fundamental source of gravity must be energy density. Crucially, this implies that the energy of the gravitational field itself contributes to the field, creating a feedback loop. Mathematically, this necessitates that the field equations must be non-linear.

**(5) The Stress-Energy Tensor:** Einstein identified Laue's stress-energy tensor ($T_{ij}$) as the energy density. Its components provide a complete physical description of the source matter: $T^{00}$: The energy density (mass-energy). $T^{0i}$: The momentum density (and energy flux). $T^{ij}$: The flux of momentum, representing pressure (where $i = j$) and shear stress (where $i \neq j$).

**(6) Generalized Poisson Equation:** In Newtonian gravity, the potential $\phi$ is governed by Poisson's equation, $\nabla^2 \phi = 4\pi\kappa\rho$ (where $\nabla^2 \phi = 0$ in a vacuum). In the relativistic framework, Einstein sought a tensor generalization of this law. He replaced the scalar term $\nabla^2 \phi$ with the curvature tensor $R_{ik}$ and the mass density $\rho$ with the stress-energy tensor term $-\kappa T_{ik}$.

**(7) The Newtonian Limit:** This requirement demands that under specific conditions—weak gravitational fields, slow motion ($v \ll c$), and static fields—General Relativity must simplify to match Newton's laws. In this limit, spacetime curvature becomes negligible and geodesics reduce to Newtonian trajectories ($F = ma$).

