# OpenReview forum: "Position: LLMs can't jump"
_ICML.cc/2026/Position_Paper_Track — ICML 2026 Position Paper Track regular_

### Official Review · Reviewer_Ldv4 · 2026-02-18

**Significance:** 2
**Argument Clarity:** 3
**Rating:** 5
**Confidence:** 3

**Questions:**

1. Do you know of any attempts to make LLMs perform abductive reasoning as outlined in this paper?
2. Do you have any ideas on how to confirm or discomfirm the ability of LLMs to perform abductive reasoning experimentally?

**Alternative Views Section:**

Yes

**Compliance With Llm Reviewing Policy A Conservative:**

Affirmed.

**Discussion Potential:**

3

**Final Justification:**

I think this paper, modulo the strong conclusion is likely to stimulate some interesting debate. The rebuttal indicates the authors are willing to moderate the conclusion which addresses my main concern with this paper.

**Paper Summary:**

The authors propose that in the absence of a consistent world model that encodes physics, LLMs are incapable performing abductive reasoning: taking presently known rules, an observed phenomena (but usually not large numbers of observations) that may contradict the known rules, and proposing new rules which explain the observed phenomena in the simplest way possible. To illustrate this they take the example of Einstein's discovery of special and general relativity, where abductive reasoning, especially counterfactual thought experiments, was required to propose new axioms which could explain the advance of the perihelion.

They argue that present AI systems are only capable of deduction: combining axioms to deduce new true statements, and induction: learning from statistical regularity.

**Position:**

Yes

**Position In Title:**

No

**Related Work:**

2

**Strengths And Weaknesses:**

**Strengths**
- I enjoyed the framing of this paper, using a known example of abductive reasoning to introduce the concepts required to understand the argument made reading this a pleasure.
- Given recent excitement about Google's Genie model and world models in general I feel that this position paper is timely and will generate interesting discussion.

**Weaknesses**

- I feel that the conclusion is too strong: The authors claim that "Our analysis confirms that the critical bottleneck is this intuitive Jump from sensory experience to formal axioms (E →A)." While they've made a well reasoned argument I don't believe this paper in anyway confirms that LLMs are incapable of abductive reasoning and further testing is required.
- The title, while funny, doesn't clearly state the position: it is unclear without further reading what "jump" refers to in this context.

**Support:**

2

---

> ### Author Rebuttal · Authors · 2026-03-31
>
> We would like to thank the reviewer for recommending to accept our paper. We are also happy to hear they enjoyed reading it.
>
> 1. Softening the Conclusion Thank you for this constructive feedback. We completely agree that the phrase "Our analysis confirms" is too strong for a position paper. We will revise the conclusion to state that our analysis "suggests" this bottleneck, explicitly clarifying that the paper establishes a theoretical framework intended to guide future empirical testing rather than claiming absolute proof.
>
> 2. Clarifying the Title We are glad you appreciated the humor! The phrase "LLMs Can't Jump" was chosen as a deliberate double entendre: it refers both to the model's inability to make a creative conceptual 'leap', and quite literally to its lack of a physical body (and thus, embodied sensory experience). While we acknowledge that the specific meaning only becomes fully clear upon reading the abstract, we hope the title's playfulness serves as an effective hook to invite the wider community into the discussion.
> Response to
>
> Q1: Prior attempts at LLM abductive reasoning Standard abductive reasoning—inferring the most likely explanation for an observation—has been thoroughly studied in the context of LLMs, most notably in domains like medical diagnosis. However, as our paper highlights, manipulative abduction is fundamentally different; in Einstein’s case, he reasoned about physical sensations that did not previously exist in the world of language.
>
> Additionally, we recently became aware of a concurrent paper on abductive reasoning (https://arxiv.org/pdf/2509.23004). They develop a neuro-symbolic system (non-LLM) that is presented with empirical facts and must successfully abduce missing axioms to verify them. Because it is not an LLM, it neatly bypasses issues of data contamination, offering an interesting alternative approach to the problem.
>
> Q2: Experimental confirmation While the implicit, 'sensual' knowledge Einstein utilized is inherently difficult to synthesize, one could approximate it experimentally by forcing an LLM to reason about novel, seemingly unrelated facts in an interactive environment.

---

> > ### Author Rebuttal · Reviewer_Ldv4 · 2026-04-02
> >
> > Thank you for your thoughtful rebuttal.
> >
> > My main concern was with the strength of the conclusion and as the revised manuscript addresses this I will raise my score.

---

### Official Review · Reviewer_vzJN · 2026-03-11

**Significance:** 3
**Argument Clarity:** 3
**Rating:** 5
**Confidence:** 3

**Questions:**

See Weaknesses.

**Alternative Views Section:**

Yes

**Compliance With Llm Reviewing Policy A Conservative:**

Affirmed.

**Discussion Potential:**

3

**Final Justification:**

I have changed my score in response to the rebuttal.

**Paper Summary:**

The paper argues that modern LLMs are structurally incapable of performing an abductive 'jump' from sensory experience to axioms , which they argue is critical for revolutionary physics discovery. Even though LLMs excel in inductive and deductive tasks, this deficiency in performing an abductive jump is a critical limitation. They argue this in detail citing the discovery of General Relativity by Einstein, and the abductive jump process through which Einstein reached at the Equivalence principle, which a modern LLM would not be able to do. Finally, they posit that enabling this jump requires moving beyond better language processing, and suggests alternatives such as world models.

**Position:**

Yes

**Position In Title:**

Yes

**Related Work:**

3

**Strengths And Weaknesses:**

Strengths:

1) The paper keeps the position discussion grounded in a concrete example of the discovery of General Relativity by Einstein. This works well for a clear position formulation and articulation.

2) Taking example from Einstein's insight "acceleration mimics gravity" to argue that this is the kind of jump that LLMs are currently incapable of doing is intuitive and well formulated.

3) The position is balanced in that it both attributes the strengths (induction, deduction) and limitations of LLMs. This puts the position in the right context within the broader discussion framework.

Weaknesses:

1) For the broader community discussion, an example which is, in some ways easier to understand than Einstein's general relativity might be better. Could the authors give an example outside of the complex GR scenario, or in a different domain.. to illustrate a discovery which would not have been possible by LLMs, but was otherwise achieved through a major abductive insight?

2) I understand the intuition that world models may learn richer latent structure about the environment than LLMs. However, it remains unclear why this would be sufficient for the kind of abductive conceptual leap discussed in the paper. Can the authors strengthen the argument by being more concrete as to how learning such latent representations might truly translate into new hypothesis formation, rather than improved prediction or planning alone.

For example, it would help if the authors could give some intuition for why a world model would be better positioned to discover the equivalence principle, whereas an LLM could not, in a deeper sense. Simply learning some latent structure does not necessarily mean the model can use it in a meaningful way like Einstein did.

**Support:**

3

---

> ### Author Rebuttal · Authors · 2026-03-31
>
> We thank the reviewer for their constructive feedback and for identifying our position as "intuitive and well-formulated." We address the two primary weaknesses below:
>
> 1. Simpler example of abductive insight
> we agree that a more accessible example would benefit the broader community. In the revised manuscript, we will include Archimedes’ "eureka" moment as a foundational example of manipulative abduction. Archimedes needed to determine if a crown was pure gold without damaging it. The solution—water displacement—was not reached via statistical analysis (induction) or derived from existing physical axioms of the time (deduction). The abductive jump: sensing the buoyancy of his own body in a bath. This sensory "simulation" allowed him to abduct a new rule: that the volume of displaced water equals the volume of the submerged object.
>
> The LLM gap: While an LLM can describe "Archimedes' principle" because it exists in its training corpora, it lacks the sensory grounding to originate such a principle from raw physical experience. It functions as a "Chinese room," manipulating symbols of buoyancy without the "feeling" of displacement that serves as the cognitive bridge to the axiom.
>
> 2. Why world models enable hypothesis formation (prediction vs. invention)
> We appreciate the push for deeper intuition on how latent representations translate into new hypotheses. We will expand section 4 to clarify this using counterfactual intervention.
>
> Beyond passive prediction: current video generation models (e.g., veo) predict the "most likely" next frame based on correlation, which remains a form of induction. In contrast, action-controllable world models (e.g., genie) allow an agent to actively intervene in a simulation.
>
> The "physical prior": a world model grounded in physical consistency provides a "synthetic laboratory." In Einstein’s case, the equivalence principle was linguistically unstated, making its symbolic probability in an LLM near zero. However, in a world model, an agent can "feel" the indistinguishability of acceleration and gravity through internal manipulation.
>
> Invention as verification: We argue that world models increase the generation probability of novel axioms by providing a substrate for verification via simulation. An agent can conjecture a principle in language and immediately test its consistency against its internal physical priors. this bypasses the need for external experimental data—which, as in the case of general relativity, may not be available for years or decades after the theoretical "jump."

---

> > ### Author Rebuttal · Reviewer_vzJN · 2026-04-04
> >
> > Questions are resolved. I like the Archimedes example. I will change my score to reflect this.

---

### Official Review · Reviewer_m94K · 2026-03-13

**Significance:** 3
**Argument Clarity:** 4
**Rating:** 4
**Confidence:** 3

**Questions:**

No additional questions.

**Alternative Views Section:**

No

**Compliance With Llm Reviewing Policy A Conservative:**

Affirmed.

**Discussion Potential:**

4

**Final Justification:**

Rebuttal has addressed my concern so I will keep my rate.

**Paper Summary:**

This paper makes a position argument that current frontier LLMs lack a fundamental reasoning mechanism of abduction. It uses the process that Albert Einstein derives General Relativity as a contrasting example to show the gap existed in current LLMs for reasoning.

**Position:**

Yes

**Position In Title:**

Yes

**Related Work:**

3

**Strengths And Weaknesses:**

Strengths:
- This paper gives a very clear and systematic explanation and classification of different reasoning abilities, and reveals abduction as a fundamentally lacking reasoning capability. It is very insightful and certainly benefits researchers in machine learning community to reconsider how to benchmark reasoning capabilities of LLMs in the future.
- Using history that Albert Einstein derives General Relativity is an impressive and persuasive examples supporting the importance of abduction.
- This paper's writing is generally good and well-organized.

Weaknesses:
- Discussions of alternative views or opposite opinions (e.g., frontier LLMs already have abduction capability but need some prompt/context engineering to stimulate it) are needed.
- A minor issue: please follow ICML submission template to have line numbers in the paper.

**Support:**

3

---

> ### Author Rebuttal · Authors · 2026-03-31
>
> We would like to thank the reviewer for recommending to accept our paper, and for finding it insightful for the ML community.
>
> 1. Discussion of alternative views (e.g., prompting/context engineering). Thank you for this excellent suggestion. We agree that the paper would benefit from addressing advanced prompting techniques. In the revised "Alternative Views" section, we will add a paragraph discussing prompting techniques that could improve abductive reasoning. We will clarify that while these techniques drastically improve an LLM's reasoning and search capabilities by breaking down problems, they still operate entirely within the existing symbolic space. Prompt engineering can help an LLM find a clever combination of existing concepts, but it does not provide the sensory grounding required for manipulative abduction—the generation of entirely new axioms without symbolic precedent.
> Please also see our comment to R Ldv4 regarding existing work on abductive reasoning.
>
> 2. Missing line numbers. We apologize for this oversight and will ensure strict adherence to the ICML formatting guidelines, including line numbers, in the camera-ready version.

---

> > ### Author Rebuttal · Reviewer_m94K · 2026-04-03
> >
> > My concerns have been addressed so I will keep my rate.

---

### Decision · Program_Chairs · 2026-04-30

**Decision:**

Accept (regular)

**Comment:**

This paper was a fun read given that the authors grounded their argument in a compelling example (the process through which Einstein developed the General Theory of Relativity). All reviewers enjoyed the framing and felt the argument was well thought out and illustrated through the example. Final ratings after rebuttal were uniformly positive: m94K (4: Borderline accept), vzJN (5: Accept, raised), and Ldv4 (5: Accept, raised).

Reviewer m94K wanted discussion of whether prompting or context engineering might unlock abductive capabilities. The authors clarified in rebuttal that while such techniques improve reasoning within existing symbolic space, they cannot provide the sensory grounding required for manipulative abduction—generating axioms without symbolic precedent.

Reviewer vzJN wanted an additional example that could be easier for the reader to digest and link to the authors' argument. In the rebuttal the authors offered to include Archimedes' principle, which satisfied the reviewer due to its simpler construction. This reviewer was also unsure about the claim that world models are sufficient for abductive leaps, but the authors also addressed this concern in their rebuttal satisfactorily—the world model creates a synthetic laboratory to test counterfactuals/reasoning that goes beyond the standard linguistic data.
Reviewer Ldv4 was concerned that the conclusion was too strong (that LLMs can't do abductive reasoning because they lack world models). The authors handled this concern in the rebuttal by committing to revise language from "confirms" to "suggests," framing the paper as establishing a theoretical framework for future empirical testing rather than claiming absolute proof.

Reviewers independently converged on similar concerns: strength of claims, need for accessible examples. The authors addressed all substantively in rebuttal. Given the unanimous positive reviews, and my own assessment that the argument is interesting and persuasive, I recommend this paper for acceptance. I think as a position piece it is thought provoking. The authors have committed to adding the Archimedes example and expanding Section 4's discussion of how world models enable hypothesis formation; I expect these revisions to strengthen the final version.